# Assisted Suicide and Euthanasia in Mental Disorders: Ethical Positions in the Debate between Proportionality, Dignity, and the Right to Die

**DOI:** 10.3390/healthcare11101470

**Published:** 2023-05-18

**Authors:** Matteo Scopetti, Donato Morena, Martina Padovano, Federico Manetti, Nicola Di Fazio, Giuseppe Delogu, Stefano Ferracuti, Paola Frati, Vittorio Fineschi

**Affiliations:** 1Department of Medical Surgical Sciences and Translational Medicine, Sapienza University of Rome, 00189 Rome, Italy; 2Department of Anatomical, Histological, Forensic and Orthopaedic Sciences, Sapienza University of Rome, 00185 Rome, Italy; 3Department of Human Neuroscience, Sapienza University of Rome, 00185 Rome, Italy

**Keywords:** euthanasia, assisted suicide, mental disorders, proportionality, dignity, right to die

## Abstract

The admission of people suffering from psychiatric and neurocognitive disorders to euthanasia and physician-assisted suicide (E/PAS) in some European and non-European countries represents a controversial issue. In some countries, the initial limitation of E/PAS to cases of severe physical illness with poor prognosis in the short term has been overcome, as it was considered discriminatory; thus, E/PAS has also been made available to subjects suffering from mental disorders. This decision has raised significant ethical questions regarding the capacity and freedom of self-determination; the family, social, and economic contexts; the social consideration of the sense of dignity and the pressure on the judgment of one’s personal value; the contextual therapeutic possibilities; the identification of figures involved in the validation and application; as well as the epistemological definitions of the clinical conditions in question. To these issues must be added the situation of legislative vacuum peculiar to different countries and the widespread lack of effective evaluation and control systems. Nonetheless, pessimistic indicators on global health status, availability of care and assistance, aging demographics, and socioeconomic levels suggest that there may be further pressure toward the expansion of such requests. The present paper aims to trace an international overview with the aim of providing ethical support to the debate on the matter. Precisely, the goal is the delimitation of foundations for clinical practice in the complex field of psychiatry between the recognition of the irreversibility of the disease, assessment of the state of physical and mental suffering, as well as the possibility of adopting free and informed choices.

## 1. Introduction

A growing number of countries in Europe and around the world have admitted the possibility for citizens to access modalities of life cessation [1]. The two main ways of implementing this are euthanasia (E) and physician-assisted suicide (PAS). The substantial difference between the two practices consists of the acting subject. In euthanasia, it is the health professional who generally administers a lethal drug (for example an injection or infusion of a substance); in assisted suicide, on the other hand, the lethal drug is prepared by the health professional and deliberately taken by the person, possibly with the aid of machines in case of reduced physical capacity. Overall, euthanasia and physician-assisted suicide (E/PAS) can be summarized in the definition of “assisted dying” [2]. Advances in medicine and the prolongation of life expectancy have raised problems previously hidden by a general foreclosure towards suicide [3]. In recent decades, medical practice has substantially overcome the paternalistic model, recognizing the patients’ autonomy [4], and E/PAS has been introduced in the healthcare sector in several countries. In this regard, it is worth noting that various factors, such as religious beliefs, demographic, social, educational, and economic factors, as well as the levels of “permissiveness” of the countries, have contributed to its acceptance [5]. These issues have influenced legislative initiatives, as significant peculiarities exist in the different parts of the world [6].

Generally, legal systems presuppose that the choice for E/PAS is made by a competent person with a non-terminal condition resistant to treatment, or untreatable, causing unbearable suffering [7,8,9].

In most legislations, the applicant must have a conscious, free, clearly affirmed, and verified will to be admitted to E/PAS practices. Furthermore, external pressure or personal advantages for others are forbidden. As reported by the Italian National Bioethics Committee, the legal admissibility of life cessation practices has been extended in several countries not only to terminal physical illnesses but also to physical and mental medical conditions that cause enduring and unbearable suffering, for which reasonable therapeutic alternatives are absent [10]. This is the case with some long-standing legislations (e.g., Benelux) and recent progressive legislative changes (e.g., Canada). Furthermore, the legislative limits that restricted access to assisted dying have been progressively challenged by requests of E/PAS by people suffering from mental disorders other than “severe mental illness” (SMI) (e.g., prolonged grief, alexithymia, factitious disorder, dissociative disorder, reactive attachment disorder, kleptomania); where SMI, according to the Dutch consensus definition, should be represented by a long-lasting psychiatric disorder which “requires psychiatric treatment” and is “accompanied by serious restrictions in the social and societal functioning” [11]. In this sense, psychic pain, a transdiagnostic dimension, seems to have an important role [12], encompassing different types of suffering (e.g., psychological, existential, spiritual) [13].

The “slippery slope” metaphor has been proposed to describe the condition for which initially excluded cases are progressively admitted in practice, both through legislative changes and jurisprudential innovations on the subject.

The present paper aims to trace an international overview to provide ethical support to the debate on the matter. Precisely, the goal is the delimitation of foundations for clinical practice in the complex field of mental health between the recognition of the irreversibility of the disease, assessment of the state of physical and mental suffering, as well as the possibility of adopting free and informed choices.

## 2. International Regulatory Framework

Since the Dutch experience, dating back almost 30 years, several parliaments have passed laws in recent years to guarantee assistance at the end-of-life [14]. As regards euthanasia and assisted suicide, to date, the countries in which E/PAS practices are allowed are the so-called Benelux block (Belgium, the Netherlands, and Luxembourg), Canada, and Spain. Another interesting case for the applicability of end-of-life to psychiatric and neurocognitive disorders is represented by Swiss legislation, where only PAS is legal (Table 1).

The main regulatory experiences in Europe are undoubtedly represented by the laws on E/PAS approved in the Netherlands with the “Termination of Life on Request and Assisted Suicide (Review Procedures)” [14] and in Belgium with the “Belgian Act on Euthanasia” [15], two texts that became effective in 2002, both having—inter alia—unbearable and irremediable physical and/or psychological suffering as criteria for admissibility.

In Dutch legislation, while both euthanasia and support for suicide remain essentially two crimes, with the approval of the “Euthanasia Act” the exemption from criminal liability for health professionals who assist E/PAS has been provided.

Interestingly, the regulatory act provides clarifications for non-competent subjects. Specifically, this could be the case for subjects suffering from neurocognitive disorders. According to the most recent document, the “Euthanasia Code 2018”, the decision-making competence to request E/PAS remains valid in the initial stages of cognitive impairment. It is properly in these stages that a person could develop profound suffering along with the awareness of having an irreversible condition leading to a progressive reduction of cognitive and functional abilities, together with the fear of depending more and more on others. Concerning the advanced stages of dementia, however, the possibility of accessing E/PAS is still provided as long as the person expressed a free choice in this sense *ex-ante*, when fully competent. The advance directives must be validated by the physicians involved in the decision-making process, based on the entire course of the disease, the subject’s behavior, the residual communication methods, the absence of contraindications, and the persistence of unbearable suffering.

For patients with psychiatric disorders, precise indications are contained in the “Euthanasia Code 2018”. First of all, the patient should be fully aware of the disorder, and it is necessary to exclude a reduction of his/her ability to judge, thus ensuring decision-making competence and the voluntary and well-considered nature of the request. The absence of any prospect of improvement and of a reasonable alternative is also required.

The refusal of a therapeutic possibility excludes the definition of “suffering with no prospect of improvement”. The latter assessment, however, jointly includes the physician and the patient, who must consider any treatment as a realistic alternative to alleviate or end suffering, notwithstanding that no patient can be forced to undergo any possible form of treatment.

In Belgium, the “Belgian Act on Euthanasia” provides, as in the case of the Dutch statute, a series of elements that, if respected, exclude the punishment of the healthcare professional. It substantially reflects the Dutch Code, although with some more articulated points and, at the same time, more controversial. For example, the possibility of accessing E/PAS is also provided for emancipated minors, a condition that has not been appropriately defined, especially in anticipation of the possibility of requesting E/PAS for mental disorders (pE/PAS) [16]. This criterion was amended in 2014, with the extension of the possibility of accessing E/PAS for all minors judged to possess the capacity for discernment, in compliance with all the other criteria provided, and only for physical diseases, even if the provisions for emancipated minors have not been changed [17].

Additionally, two points are particularly controversial:-no specification is provided on previously attempted treatments required to define the medical condition as “cannot be alleviated”;-the concept of the “medically futile condition”, as the situation in which the patient must be, seems epistemologically fragile, devoid of quantitative validity, and precarious also from a qualitative point of view, as emerged by comparing the opinions between groups of doctors, nurses, and patients [18,19]. Moreover, although the connotation of futility has ethical foundations for some organic disease treatments, it could be difficult to apply to mental, especially psychiatric, disorders.

Moreover, even Luxembourg, with its “Euthanasia and Assisted Suicide Act” of 2009, allows E/PAS.

Spain has recently joined the Benelux countries’ legislations, where E/PAS is also extended to patients with mental disorders, with the “Ley Orgánica 3/2021, de 24 de marzo, de regulación de la eutanasia”. However, unlike the three aforementioned countries, Spain has integrated into its legal system not decriminalization but the legitimacy of both euthanasia and medically assisted suicide, procedures included in the constitutionally guaranteed fundamental rights of the person.

Since 1942, Switzerland has allowed PAS, as long as the motives are not selfish.

In general, the extension of assistance in dying to people with diagnoses other than organic disorders has often been supported by the alleged discriminatory connotation of the differentiation between the latter and mental disorders, assumed instead as an equal cause of unbearable suffering [20].

A similar extension also occurred in Canada, where E/PAS was introduced in 2016 for both physical and psychological suffering conditions but with the limitation connected to the necessary presence of diseases such that natural death is “reasonably foreseeable”. In February 2020, an amendment to the law was proposed aimed at eliminating this constraint and making E/PAS accessible even to those suffering from non-terminal illnesses, therefore also those of a psychiatric nature. In 2021, this change became effective with the approval of a new law on medical assistance for dying. In the new legislation, the “reasonable foreseeability of natural death” criterion was repealed; accessibility to people suffering solely from a mental illness was excluded until 2024, pending the elaboration of specific protocols.

## 3. Data on Euthanasia and Assisted Suicide

A key element in the statistics on E/PAS is the constant increase in the number of people who have requested and obtained access in countries where such practices have been introduced [21]. Stable or increasing numbers of E/PAS cases, although numerically lower, were also recorded for people diagnosed with mental disorders (included in the classifications of mental disorders, such as DSM-5 and ICD-11), i.e., psychiatric, neurocognitive, and neurodevelopmental diseases (pE/PAS). The main epidemiological data come from European countries such as the Netherlands, Belgium, and Switzerland, where for some years E/PAS has been introduced as a care practice.

### 3.1. Netherlands

For the Netherlands, data are collected from two large E/PAS control and evaluation centers, the Dutch Regional Euthanasia Review Committees (DRERC, Regionale Toetsingscommissies Euthanasie) and the End-of-Life Clinic (now renamed Expertisecentrum Euthanasie); the latter represents an organization that refers people, who have not received the consent of physicians to their requests, to enter E/PAS procedures. DRERC data for pE/PAS show how the number of cases per year has grown from 0 (2009) to 88 (2020) [22].

A similar trend was also recorded by the End-of-Life Clinic between 2013 and 2017, with an increase from 9 to 65 cases [23].

Interesting data comes also from a survey that involved Dutch psychiatrists, who reported that psychiatric patients who had explicitly requested access to pE/PAS between 2015 and 2016 were 1100–1150, and pE/PAS was provided for around 60 to 70 patients [24].

For pE/PAS, it was possible to observe that more than two-thirds of the cases concern women: a review conducted on pE/PAS cases in the Netherlands between 2011 and 2014 revealed a higher prevalence of women (46 out of 66 cases; 70%) [25], with depression (36; 55%) as the main psychiatric condition.

Moreover, depressive symptoms (7%) and tiredness of living (17%) are among the most frequent reasons given for all kinds of E/PAS requests [26].

Reviews of pE/PAS cases also reveal an important prevalence of personality disorders (PDs) [27]. Recent data from the DRERC highlight that, between 2011 and 2017, pE/PAS was provided in the Netherlands to 74 people with possible PDs, of which 65% had definite diagnoses [28]. The female gender was the most represented (76%) and cluster B, as expected, was the most frequent (in about two-thirds of cases); both psychiatric and organic comorbidities were frequent and present in 97% and 62% of cases, respectively. The most frequent comorbid disorders were unipolar or bipolar depression (70%), post-traumatic stress disorder (PTSD) or post-traumatic symptoms (31%), anxiety disorders (31%), and somatoform disorder (19%). As for the therapeutic pathways, 73% of people had a history of hospitalization in psychiatric wards, while 14% had undergone compulsory treatments. Approximately one-third of the people had been given electroconvulsive therapy (ECT), while full drug treatment for depression, including monoamine oxidase inhibitor (MAO-I), was reported in only 7% of cases. Although some form of psychotherapy had been undertaken in 72% of patients, no evidence-based typologies specific to cluster B personality disorders were used. On the other hand, about half of the patients refused therapeutic proposals, whether it was hospitalization, drugs, or psychotherapy, mostly due to a lack of motivation for treatment. A role was probably also played by social isolation and loneliness, reported in more than half of the cases of patients with PDs, in addition to interpersonal conflicts and socioeconomic stress [29].

In the DRERC report for the year 2019 [30], people diagnosed with dementia suffered in almost all cases (160 out of 162) from an early stage cognitive impairment. Such patients were thus aware of their own state of health and of the initial symptoms of the disease (loss of orientation and personality changes). In an anecdotal but significant proportion of cases (2 of 162), subjects suffered from an “advanced or very advanced” clinical condition or were no longer competent. It emerged that there is a greater propensity to choose E/PAS in the range between 64 and 74 years, despite some kind of protection given by the diagnosis of dementia [31,32]. Other risk factors for the choice of E/PAS among elderly people included depression (which was no longer a risk factor after treatment), a lifestyle based on the quality of life rather than on life itself, less religiosity, higher education, and greater socio-economic status.

### 3.2. Belgium

In Belgium, the review of data from the registers of the Federal Control and Evaluation Committee on Euthanasia for the years 2003 to 2013 showed a progressive increase in the total number of people who had access to E/PAS (from 235 to 1807), with a similar trend, albeit on a smaller scale (from 0.5% of all cases in the period 2002–2007 to 3% in 2013), for patients diagnosed with psychiatric disorders or dementia [33,34]. The most recent data (2018–2021) show that the percentage of pE/PAS has remained between 1 and 2% of the total number [35]. In absolute terms, in the eleven years of observation between 2002 and 2013, 179 cases with a diagnosis of psychiatric disorder or dementia received pE/PAS. Most (83; 46.4%) had a mood disorder as the only diagnosis, while a minor part (12; 6.7%) had an additional comorbid psychiatric disorder (mainly PDs); other cases (22; 12.3%) had various mental disorders, including autism (6; 3.3%), borderline personality disorder (3; 1.7%), anorexia nervosa (3; 1.7%), and post-traumatic stress disorder (2; 1.1%). The diagnosis of dementia was also frequent, affecting 62 people (34.6%).

As from the latest report of the “Commission fédérale de contrôle et d’évaluation de l’eutanasie” for the years between 2018 and 2019 [36], the number of pE/PAS for people with psychiatric disorders (ICD-10, F10–F99) remained stable: 27 in 2016, 26 in 2017, 34 in 2018, 23 in 2019; while the number of people with cognitive impairment doubled in 2018 and 2019 (48 cases) compared to 2016–2017 (24 cases).

The data regarding pE/PAS for the period 2003–2013 reveal a prevalence of the female gender (88 cases out of 117; 75%).

Interestingly, in 11 cases (out of 48 patients admitted to pE/PAS from 100 requests) the process was interrupted after having obtained the eligibility. Mostly, this was motivated by the fact that the possibility to proceed with E/PAS had provided the relief they needed to keep on living [37]; in the follow-up performed between 2007 and 2012, 57 of the 100 people who had requested pE/PAS were still alive and, for many of them (48 cases), their requests were on hold because they were managing with regular, occasional, or no therapy. On the contrary, 6 subjects had completed the suicide autonomously. In this regard, it should be emphasized how often for patients with mental disorders the possibility to express suicidal intentions can represent a way to communicate their suffering and elicit more intense therapeutic interventions [38].

In addition, the data of the Federal Commission for the Control and Evaluation of Euthanasia for the years 2016–2017 reveal that, in over 80% of cases of subjects with terminal diseases, the main suffering was of psychic nature, such as dependence on third parties for treatment, loss of autonomy, loneliness, despair, sense of uselessness, desolation, and reduced social contacts [39].

### 3.3. Switzerland

In Switzerland, the data are incomplete as the Swiss Federal Statistical Office set up a central register of cases only in 2011; previously, data collection was the prerogative of organizations active in the field of death assistance as well as, in some cantons, forensic institutes and public health services. The exact number of real cases is therefore difficult to establish. However, a retrospective study that analyzed the data of some forensic medical institutes for the period 1985–2014 found that the number of cases of assisted suicide due to purely psychiatric diseases was 61, accounting for 2.1% of the total [40]; among these, 43 (2.9%) were residents in Switzerland and 18 (1.3%) were residents in other countries. Regarding the diagnosis of dementia, the percentage of total cases was 4.5% [41].

### 3.4. Canada

For Canada, the 2021 annual report [42] indicates a steady increase since 2016, the year in which the Canadian Parliament regulated access to E/PAS. In general, the number of people who had access to suicide aid went from 1018 cases in 2016 to 10,064 in 2021. Similarly, although not frequent in absolute terms, cases of dementia were fairly represented with a percentage of 0.9% (0.5% in 2020). On the other hand, there is a lack of data on the frequency of psychiatric disorders.

Finally, it should be emphasized that the numbers are substantially underestimated due to the non-inclusion of people with physical diseases comorbid with psychiatric disorders; this is the case for patients with terminal diseases and depression or patients with cognitive impairment and other neurological diseases.

## 4. Debated Issues

### 4.1. Controversial Aspects

Several controversial aspects emerge when considering E/PAS for people with solely psychiatric disorders or with different types of cognitive impairment.

The first aspect is given by the overlap between mental illness and suicidal ideation, as well as the free and well-considered will to access E/PAS. Psychiatric disorders represent one of the main risk factors for suicide [43]; on the other hand, the psychiatrist’s medical, social, and human role consists precisely in the treatment of psychopathology, including suicidal ideation, and in the prevention of suicide [44]. There is also an overlap between the constituent features of many psychiatric disorders and the criterion of intolerability of mental suffering. Briefly, unbearable suffering could be included in the diagnostic or severity criteria of psychiatric disorders, rather than representing the expression of a free and independent choice [45]. For example, according to the DSM-5 [46], for major depressive disorder, recurrent thoughts of death and suicidal ideation, as well as a suicide attempt or a plan to commit suicide, are considered diagnostic features (criterion 9). In borderline personality disorder (BPD), on the other hand, “recurrent suicidal behavior, gestures, or threats, or self-mutilating behavior” contribute to the diagnosis (criterion 5) [46]; aspects of emotional and behavioral dysregulation, related to an increase in suicidal risk [47], are also included as criteria. BPD patients represent an important percentage of all cases of suicide [48], with a much greater risk than the general population, especially during adolescence and early adulthood [49]. Suicide and self-injurious behaviors are also frequently reported in other PDs, especially for cluster B [50]. In narcissistic personality disorder, for example, there is a desperate need of protecting the own image of perfection by obtaining continuous external self-affirmation [51,52]; failure can trigger a strong sense of shame and humiliation from which, sometimes, only suicide is recognized as a way out. High suicidal risk is also present in antisocial personality disorder and, in general, in disorders characterized by high levels of aggressiveness [53].

As regards the criterion of “*no prospect of improvement*” of the disease, as required by Dutch legislation, for mental disorders it is extremely difficult to establish all the possible treatments that need to be considered [54]. For PDs, for example, a series of psychotherapeutic and rehabilitative interventions are able to reduce suicidal ideation and the risk of self-injurious behaviors [55,56]. However, people suffering from these disorders frequently refuse or interrupt treatment proposals; this also applies to several disorders in which awareness of the disease (insight) may be lacking, such as psychotic disorders or those affecting neurodevelopment and cognition. In this regard, the legislation considers only the treatments accepted by the patients for the determination of the treatability or not of the disorders, giving preponderance to the subjective point of view. This consideration, however, does not value some judgment-distorting aspects caused by the same disorders, such as hopelessness in depression [57] or impulsivity and cognitive distortions in BPD [58], that are related to the dimension of suicidality. Furthermore, people who attempt suicide have important cognitive distortions regardless of the degree of depression (e.g., greater rigidity, dichotomous thinking, excessive generalization, selective abstraction, irrational thoughts, hopelessness, overgeneral memory, perfectionism, and deficit in problem-solving); thus suggesting that these alterations may represent specific therapeutic goals, in addition to the treatment of any associated psychiatric disorders [59].

Given the principle of the centrality of the patient, with their subjectivity and right to self-determination, it must be considered that cognitive distortions, hopelessness, and helplessness can represent the symptoms of a psychiatric disorder, and, at the same time, influence the choices of patients and generate mistrust towards the efficacy of the proposed treatment(s). At the same time, similar aspects may also affect patients’ social context, including family members and health professionals, especially in the most complex and frustrating situations [60]. This could generate a vicious circle in which the discouragement of the surrounding support system adds to the patients’ uncertainty, reinforcing their belief that E/PAS is the only viable path [61].

A further problematic aspect is that E/PAS could represent a lethal means for people with a psychiatric disorder and suicidal intentions, especially for the female gender [62]. This risk is highlighted by an overlap between the high prevalence of women for cases of E/PAS requests and admissions, and suicide attempts [63]. On the contrary, men are more prone to adopt effective means to commit suicide [64], as well as being less likely to seek professional support for mental suffering [65]. Data from North America confirm that women, especially those of European descent, account for about half of all cases of medically assisted suicide, while they only represent a minority of cases of suicides [66]. The risk of providing a more lethal means is also increased by some debates and proposals relating to the possibility of allowing elderly people who feel “tired of living” to access E/PAS, even in the absence of physical or mental diseases that can justify the request [67,68].

### 4.2. The Slippery Slope

The metaphor of the “slippery slope” has been formulated in order to describe the expansion of the conditions of eligibility to E/PAS, which led to the admission of cases initially excluded through legislative regulations [69,70], as in the case of Canada. However, the metaphor can also be applied when referring to daily practice, where more and more requests for E/PAS can be considered valid; this is a risk defined by some authors as the “normalization” of E/PAS [71]. Furthermore, any restriction on access to E/PAS can be considered discriminatory as they may constitute a violation of the right to autonomy and self-determination; on the other hand, E/PAS could gradually be considered, even among healthcare personnel, an increasingly valid and ethically correct solution for respecting the autonomy and dignity of patients. Although the increasing number of people accessing E/PAS supports these risks, studies in the Netherlands and Belgium reveal that this increase reflects the reduction of cases of “administration of lethal drugs without explicit patient consent”, suggesting emancipation from illicit practices [72].

However, the risk of the “slippery slope” persists, especially considering the extension of E/PAS for people with mental disorders. For example, it is known how news broadcasting about suicide cases and the means used can increase the risk of emulation (also known as the “Werther Effect”), particularly among the most vulnerable subjects with psychopathological and social problems [73].

Another risk, especially for more complex cases, is the reconsideration of E/PAS as a solution to the lack of adequate care systems to meet patients’ needs. There is evidence that psychiatric patients need an improvement in the accessibility and quality of mental healthcare, as well as a profound change in society’s perception and support for them [74]. This problem deserves particular consideration, given the low investment rates in mental health policies. In this regard, the WHO report is extremely worrying [75]: in low- and middle-income nations, less than 1–2% of the health budget is devoted to the prevention and treatment of mental disorders. On the other hand, most of these funds are intended for the maintenance of long-term care facilities where the rate of psycho-socio-functional recovery of people is very low, with prevailing isolation, violation of human rights, and deterioration of health status. Furthermore, the WHO reiterates the poor governance of services and the difficulties in planning interventions due to the lack of integration of welfare services and the different local agencies; this affects the difficulties of patients, family members, and the economic system itself, in terms of loss of productivity and purchasing power. In historical periods plagued by viral pandemics [76,77], wars, and growing economic disparities, it is difficult to avoid thinking that the introduction of E/PAS cannot become a simple solution to complex problems.

Even the concepts of human dignity and dying with dignity can be undermined by social pressures that enhance the aspects of performance, autonomy, and independence, in an unbalanced manner that could elicit ambivalence in patients [78]. Such connotation of dignity, in fact, is based on the performance aspects of the subject who holds it temporarily, depending on the situation [79]. This definition of dignity, based on inter-subjective perspectives, could also pose the risk of stigma against people with mental disorders, who can internalize stereotypes and prejudices, thus limiting judgment on their sense of dignity. From a Kantian perspective, the concept of human dignity should be based on an internal and individual perception, or instead on self-determination and decision-making in relation to one’s goals, desires, needs, and impulses [80]. Therefore, it is possible to affirm that the conditions that limit or deprive human beings of the possibility of self-determination and of living according to their own moral identity are contrary to their sense of dignity.

Therefore, in preventing the risk of the so-called “slippery slope”, it is necessary to consider the health conditions that effectively and irreversibly deprive people of the possibility to free determination, by distinguishing them from cases in which subjective (physical or mental suffering) and objective (lack of means) conditions may still be modifiable.

### 4.3. The Role of Health Professionals

Surveys of healthcare professionals’ opinions on E/PAS usually reveal lower support than the general population and a limited propensity to have a formal role in these procedures [81]. In a recent survey on the support for E/PAS in people with psychiatric disorders, discrepancies were highlighted between the acceptance and conceivability of the general public (53%) and that of health professionals, as well as within the latter group [82]. While a fair number (47%) of general practitioners declared support, the percentage fell among psychiatrists (39%) and medical specialists (20%). Female gender, religious beliefs, specialization, and doubts about possible improvement perspectives (the latter only for psychiatrists) were related to lower conceivability. In contrast, previous experiences with pE/PAS practices were associated with increased acceptability.

A critical element for psychiatrists was raised by doubts about the potential for clinical improvement [83]: more than 65% of them claimed that it was possible to determine whether the suffering of a psychiatric patient was unbearable and without prospects of recovery. A slightly lower percentage (64.2%) considered assistance to suicide as part of healthcare.

A further study surveyed the physicians’ and public attitudes toward euthanasia in people with advanced dementia, as requested by written advance directives [84].

A major difference emerged again between the opinion of the general population (60% was favorable) and that of health professionals with 24% favorable among general practitioners, 23% among clinical specialists, and 8% among nursing home doctors.

A low rate of acceptability was registered among nursing home physicians, that is, among those most involved in the care of patients with advanced dementia. It has been hypothesized that this attitude may depend on the awareness of the complexity of the cases, the impossibility of obtaining valid consent, as well as on the knowledge of other interventions to alleviate suffering.

The practice of E/PAS is also related to strong emotional distress for the healthcare staff involved [85], both for the participation in the procedures and for the external pressure, for or against the acceptance of requests, exerted by the patients themselves, family members, or society. On the other hand, there is a growing trend among the general population regarding the acceptance of E/PAS for patients with mental disorders [86].

Although some authors consider the patient’s point of view to be a priority in the end-of-life discussion [87], it is still necessary to focus on the critical issues affecting the practitioners. This is the case, for example, with some legal disputes involving healthcare professionals who have participated in E/PAS for patients with psychiatric or neurocognitive disorders. One case concerned three Belgian physicians charged with the death of a patient who had been euthanized in 2010. They were acquitted in 2020 after several years of trial. The 38-year-old patient, according to family members, required E/PAS after the failure of a relationship; for the three physicians, she would have suffered from BPD and an autism spectrum disorder (diagnosed seven weeks before euthanasia). A further trial, known as the “Dormicum case”, involved a Dutch physician—later acquitted in 2020—accused of euthanizing, in 2016, a 74-year-old woman with late-stage Alzheimer’s disease, without properly acquiring her consent [88]. The dispute arose from the fact that the woman had drawn up two wills, the first about 4 years before the time of euthanasia (immediately after receiving the diagnosis of Alzheimer’s, justifying the request with the trauma suffered for her mother’s long stay in a nursing home due to the same disease), while the second about 1 year before the moment of euthanasia (affirming the will to decide independently when to interrupt her life). After one year, at the request of her husband, the doctor practiced euthanasia relying on the first will, stating that the patient had lost the ability to express valid consent.

The procedural steps for E/PAS raise several questions and contradictions about the evaluation method, the criteria for interpreting the patient’s will, and the final decision. Health professionals are called upon to interpret the behaviors and statements of patients both during the illness stages and in the moments preceding E/PAS. Therefore, it is essential to clarify the value to be attributed to subsequent statements that prove to be contrary to the advance directives. However, it should be established how and to what extent the person is deemed to be able to express valid consent; it is appropriate to determine which expressions, signs, and behavior should be valued for manifesting will or unbearable suffering. A further controversy arises from the possibility of concretizing an advance directive for E/PAS at a time when the person has a significant reduction or no decision-making ability: the fact that the doctor can be given the task of clearly distinguishing the judgment on these two dimensions, namely the will to die and the intolerability of suffering, can be disputed. In summary, a subjective psychological condition—suffering—is assimilated to an objective datum—the situation—for which an advance directive was made, and then shifts back to the subjectivity of a third party—the doctor called to judge it—with an evident confusion of roles. Finally, it must be established whether E/PAS is a medical act or a crime that is amended for certain circumstances; in the first case, precise assistance paths and guidelines must be codified [89].

### 4.4. Cultural Issues

In Western societies, the tradition of euthanasia derives, as its term suggests (from the Greek *eu*, “good,” and *thanatos*, “death”), from the civilization of ancient Greece [90]. The ancient Greeks perceived the debilitation of old age as humiliating, which placed limits on the expression of their life and the pursuit of posthumous fame. They also abhorred anything that went beyond the canons of symmetry and balance [91]. In general, they had a positive attitude towards the “good death”, a definition that appears for the first time in *Myrmiki*, the last comedy by Posidippus (ca. 300 BCE), where it is considered the best gift of the gods. In the Epicurean philosophical system, pain was to be avoided, as was the fear of death. Plato, in some way also referring to the traditions of Sparta, was in favor of “passive euthanasia” (“Republic”, ca. 374 BCE), as individuals sick in body and soul should be abandoned to die for the sustainability of the city. On the contrary, Hippocrates, in his Oath (400 BCE), affirmed the prohibition for doctors to “prescribe a deadly drug to please someone, nor give advice that may cause his death”. The subsequent permeation of European societies by the increase in Judeo-Christian teachings has progressively led to a proscription on suicide and euthanasia: since life is a holy gift, the decision on the moment of death belongs exclusively to God.

This view changed during the Renaissance and the Enlightenment, when it was stated that people have the right to decide how to die [92]. In the modern era, the progressive loss of the paternalistic figure of the physicians on the one hand, and on the other the possibility of an indefinite extension of life, even in conditions of serious illness, thanks to biomedical technologies, have renovated the debate on the end-of-life. New problematic issues have been raised, and the controversial aspects of assistance for dying are still far from being resolved. The risk of the return to the paternalistic figure of the physicians has been highlighted in relation to the fact that they should decide on the cessation of life for subjects not competent to explicit consent. There is also a wide debate on the duties of doctors, for whom the professional mandate to care for patients, relieve pain, and respect their choices, in the case of E/PAS, is challenged by the possibility of ending their life [93]. The role of healthcare professionals should also be discussed concerning whether and to what extent assistance with dying should be considered a medical act. In the Swiss system, for example, physicians are involved in the certification of the patient’s mental competence, in the assessment of the congruence of the state of illness with the request, and in the prescription of the lethal drug [94]. The entire procedure is often supervised by volunteer staff and the presence of a physician is not mandatory in the preparation or administration of the drug, which often takes place in the domestic intimacy of the applicant. In this regard, an open issue concerns the fact that non-medical personnel are emotionally involved and exposed to the death of people with terminal and non-terminal diseases.

From a global view, it is worth noting that the state of the art on end-of-life, if it is already a debated topic in Western countries, finds further critical points worldwide. The vision of Islam is very similar to that of Christianity, since the body is considered a gift and a loan from Allah, who is the only effective owner; thus, it is only he who can possibly decide on the end of a person’s life. A violation of this precept is considered a sin [95]. Heterogeneous elements are instead present in the Buddhist religion. According to the Buddha’s teachings, people, by freely disposing of their own life, can also dispose of its end. However, unsolved problems will be passed on to the next life and suicide represents an obstacle on the way to enlightenment that should be avoided [96]. Similar aspects are also present in Hinduism. Although suicide is punished with diseases or other ill conditions in the next life, death by abstinence from eating or drinking for religious purposes is considered a respected ascetic practice [97].

In general, religious dictates and cultural roots based on religious traditions exert a preventive effect against voluntary death, which is particularly important with regard to the risk of suicide. However, on the other hand, it must be said that this discouraging action can also favor the tendency toward prohibition and stigmatization of the end-of-life, regardless of the specific context in which it develops [98,99,100].

About pain, there are different traditional views on the topic [101,102], which in some cultures is considered an attempt by the body to heal, a purifier of sins, or a punishment for misdeeds [103]. There are also fatalistic visions that see pain and early death as a consequence of guilt [104]. It is not surprising, therefore, that in many countries there is strong resistance and opposition to E/PAS. It should also be said that within multicultural societies, subgroups with specific attitudes make the consideration of E/PAS heterogeneous. These cultural issues should be considered in the evaluation of legislative choices on the end-of-life, particularly if the reason concerns non-terminal illnesses such as mental disorders.

### 4.5. Contrasting Values

Reasons for ‘pros’ and ‘cons’ influence the choice of legalization or decriminalization of E/PAS, as well as some position statements of medical associations. The main arguments used to support E/PAS are [105]:-ensuring respect for patient autonomy, which is one of the fundamental bioethical principles, in combination with justice, beneficence, and non-maleficence [106];-allowing relief of refractory suffering;-ensuring a safe medical practice, with requests evaluated in several procedural steps that include eligibility criteria, safeguards, and regulations in place to protect patients;-giving reassurance and serenity to many people with terminal diseases and their loved ones by providing support at the end-of-life.

On the other hand, the main arguments used to oppose E/PAS are [106]:
-the risk of the slippery slope and suicide contagion [107], as supported by some studies that found an association between the legalization of PAS and “an increased rate of total suicides relative to other states and no decrease in non-assisted suicides” [108];-respect for the patient’s autonomy can be guaranteed through laws protecting the right of competent adults to refuse any proposed medical intervention even if such a decision could be harmful; advance directives are essential to establish therapeutic choices in the event of loss of competence [109,110];-E/PAS could potentially lead society toward an attitude that considers suffering, interdependency, and the lives of disabled or terminally ill individuals in an unfavorable way;-Institutions should protect people’s lives, especially the most vulnerable ones, and not take part in their death.

According to Fontalis et al. [109], an essential argument for institutional choices is represented by the principle of “sanctity of life”, a concept often associated, although not fully equated, with religious and cultural traditions. By this principle, since life has sanctity, its value prevails over all other values.

Another issue, often debated inside medical associations, concerns the potential conflict between the principles of “beneficence”, that a physician should act in the best interest of the patient, and that of “non-maleficence”, that is, ‘first, do no harm’, ‘*primum non nocere*’, as seen discussing the Hippocratic Oath.

Worldwide, there are different positions of medical associations on E/PAS. In its Code of Medical Ethics, the American Medical Association (AMA), referring to PAS, highlight the main arguments [111].

According to the perspective of those who oppose, the physicians’ mandate should be exclusively to support and care for patients, even when a cure is not possible. On the other hand, supporting PAS, it is argued that physicians are both professionals and moral agents who should be given the opportunity to act (or refrain from acting) in accordance with the dictates of conscience. However, they should have limits, such as:-providing care in emergencies;-honoring patients’ informed decisions to refuse life-sustaining treatment;-informing patients about all relevant options for treatment, including morally objected options;-continuing to provide other ongoing care for patients or formally terminating patient-physician relationship in keeping with ethics guidance;-respecting the work and professionalism of colleagues.

Overall, however, the position of the AMA is that of opposition to physician-assisted suicide and euthanasia, as both are fundamentally incompatible with the role of the doctor as a healer; so, the physician who performs E/PAS assumes full responsibility for the act [112].

In 2021, the British Medical Association (BMA) representative body voted for modifying the policy from opposition to a change in the law on assisted dying, to a position of neutrality towards PAS (in the UK, at the moment, both euthanasia and assisted suicide are illegal under English law), confirming opposition to euthanasia.

For the Netherlands, the Royal Dutch Medical Association (KNMG) position is that any procedure of end-of-life is a last resort, to be used in cases in which patients and physicians have exhausted all options and the suffering remains refractory to any means and unbearable [113].

According to KNMG, physicians are always responsible for determining the burden and the components of patients’ suffering, regardless of its source or the way in which patients characterize it; this occurs even if the requests stem from a sense of ‘completed’ life.

However, the judgment about ‘completed’ life is one that only persons can make for themselves, and no influence or responsibility could be identified for physicians.

A similar position, as regards PAS, is held by the Swiss Academy of Medical Sciences which acknowledges that ‘in certain special cases a doctor’s personal decision to assist a dying patient to commit suicide is in accordance with his or her conscience and has to be respected’ [114].

Although there are some differences, the various policies of the medical associations remain constant on the fact that requests, to be valid, are made on the basis of a medical condition and by competent patients.

We refer to more detailed studies for further analysis of these issues [105,115].

## 5. Conclusions

From an ethical perspective, the principle of not ending someone’s life should not be absolute. There should be cases of exception in which it would be morally legitimate, and probably mandatory, to fulfill a patient’s request to be helped to die. Precisely, specific cases should be identified in which, as an extrema ratio, only assisted dying is a valid condition to guarantee the fundamental rights of the person, as:-the right to health which, although irremediably obliterated by the disease, remains the right not to persist in a state of untreatable and unbearable suffering;-the right to dignity, self-determination, and the preservation of moral identity.

The extrema ratio, on the other hand, can only be the result of a complex evaluation process that extends beyond purely medical boundaries, involving legal authorities and society. Although E/PAS should not be considered a failure in protecting a person’s health, it is necessary to reflect on how healthcare and social support are performed, concerning the personal path of the suffering individual. In the case of people with mental disorders, it would therefore be necessary to evaluate the quantity and quality of support provided to improve the functioning of the person and to maintain a decent quality of life.

The problem of allocating resources in health and social welfare should not be considered a determinant in the overall assessment for E/PAS, especially in contexts burdened by greater economic disparities. Such an implication would fuel the “slippery slope” with the concrete risk of labeling one life as less dignified than another on the basis of economic background, with a decisive and distorting prejudice.

Exploring the dimension of unbearable suffering in people with mental disorders requiring E/PAS can detect the lack or loss of social support as well as socio-economic problems. In broad consideration of mental suffering, several determinants that do not relate to mere medical or psychiatric care could be recognized, which cannot be addressed by healthcare personnel alone. In this sense, the request to be helped to die should be an alarm signal for the local health and social services. Once the reasons have been assessed, all human and economic resources should be invested to avoid reaching the judgment of irremediability of the suffering, harshened by loneliness and social isolation.

The collection of data on requests and interventions performed would also be a fundamental step in assessing the quality and efficiency of social and health services, thus enabling the implementation of any corrective mechanisms. This could lead to an improvement in the quality of care, allowing adherence to the constitutional framework of health protection not only as an individual right but as a community interest.

The interdisciplinary framework could also shift the focus of the problem from the norms to the people [74].

Among other recommendations, a homogenization of international legislation appears desirable with the aim of avoiding inequalities of treatment and limiting the so-called “tourism of death” to countries where E/PAS is allowed. Data from Switzerland, for example, show an ongoing critical increase in absolute numbers of “suicide tourists” [116]. As seen in the section concerning cultural issues, there are many heterogeneous aspects in considering end-of-life, both inter- and intra-societies. A step forward would be at least the creation of an international charter of fundamental rights.

Furthermore, for specific countries, the absence of regulatory legislation on both E/PAS and pE/PAS should also be avoided. In the countries where such legislation exists, the majority of health professionals and the general population perceive the regulation as fair, despite the divergence of views on applicability in individual cases. Perseverance in a wait-and-see attitude towards assisted dying legislation is therefore scarcely recommended, especially in a period in which pessimistic indicators on the global health status, the availability of care, the aging population, and socioeconomic levels—all burdened by the pandemic situation and the current gloomy geopolitical scenarios—suggest further pressure towards the expansion of such requests.

In conclusion, end-of-life assistance represents an issue of enormous ethical and legal significance, such as not allowing state systems to continue to ignore the urgency of promptly resorting to specific regulations. Currently, in the inertia of legislation, an increase in borderline cases may occur, defined as emergency conditions not provided for by law. Since it is precisely the possibility of predicting and preventing extreme situations that deprive them of their exceptional nature, a collective and multidisciplinary effort is, therefore, essential for the determination of regulatory rules on the matter.

## Figures and Tables

**Table 1 healthcare-11-01470-t001:** Overview of the regulatory framework about E/PAS in different countries.

Country	Criteria
**E/PAS legal**	
**Netherlands**	- voluntary and weighted request for E/PAS - unbearable suffering together with the absence of prospects for improvement - correct information to the person on his/her condition, prognosis, and options - absence of reasonable alternative solutions ascertained together with the person - at least one independent physician must be consulted to verify that the request meets the criteria - age of at least 12 years old (parents’ consent required when aged between 12 and 16 years)
**Belgium**	- adults and emancipated minors (in 2014 euthanasia legalized for children) legally competent and aware at the time of E/PAS request - request is voluntary, well-considered, repeated, and is not the result of any external pressure - medical condition of constant physical or mental suffering, unbearable and unrelievable, “resulting from a serious and incurable disorder caused by illness or accident” - when a poor prognosis is not foreseeable for a certain period, a third physician (a psychiatrist or an expert on the specific disease) must be consulted in addition to the independent physician who is always required by law
**Luxembourg**	- competent adults (children aged 16 to 18 need parental or legal guardian consent) - unbearable pain from a physical or mental condition that cannot be relieved - voluntary and repeated request
**Spain**	- competent adults - serious, chronic, and disabling condition or serious and incurable disease, causing unbearable physical or mental suffering - for a person with impaired cognitive faculties and unable to give free, voluntary, and conscious consent, the request can be admitted in case of advance directives (“testamento vital”, “voluntades anticipadas”, or equivalent legally recognized documents)
**Canada**	- competent adults - be eligible for health services funded by the federal government, or a province or territory (or during the applicable minimum period of residence or waiting period for eligibility) - have a grievous and irremediable medical condition that meets all of the following criteria: have a serious illness, disease or disability (excluding a mental illness until 17 March 2024); be in an advanced state of decline that cannot be reversed; experience unbearable physical or mental suffering from the illness, disease, disability, or state of decline, unrelievable under conditions acceptable to the person
**Only PAS legal**	
**Switzerland**	- competent adults - serious and incurable disease causing intense physical and psychological suffering - no selfish motive
**Law and year** **Netherlands**—“Termination of Life on Request and Assisted Suicide Act 2001”; **Belgium**—”Belgian Euthanasia Act 2002”; **Luxembourg**—“Euthanasia and Assisted Suicide Act 2009”; **Spain**—”Ley Orgánica 3/2021, de 24 de marzo, de regulación de la eutanasia”; **Canada**—“Medical Assistance in Dying Act 2016”; **Switzerland**—“PAS established in the criminal code 1942”	

## Data Availability

Data were collected from several sources. A literature search through PubMed, Scholar, and Scopus was performed using terms such as “euthanasia”, “assisted suicide”, “mental disorders”, “proportionality”, “dignity”, and “right to die”. The references of the selected articles were consulted to expand the research. The legislation of the countries where E/PAS was introduced and the institutional databases containing certified data were also reviewed. We also searched for newspaper articles that contained information on some cases of interest. The data supporting the study’s findings are available from the corresponding author upon reasonable request.

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
