# Peer review of "Assisted Suicide and Euthanasia in Mental Disorders: Ethical Positions in the Debate between Proportionality, Dignity, and the Right to Die"

_healthcare, 2023, doi:10.3390/healthcare11101470_

Round 1

Reviewer 1 Report (Previous Reviewer 2)

The authors have addressed my comments and I am satisfied with the improvements.

Author Response

Thank you to the reviewer. We are also very grateful for the appreciation of our study and for the useful suggestions.

Best regards, sincerely.

Reviewer 2 Report (New Reviewer)

Thank you for the opportunity to review this manuscript. The content is very timely and relevant to an international audience. 

I suggest that you revise table 1 so it is more readable meaning a horizontal table with the main issues highlighted so the differences in the national legislation are easier to get an overview over. References to the local legislation could be inserted as footnotes. 

I think the manuscript could benefit from discussing the relevant main arguments from those countries who does not accept E/PAS. 

Additionally a link to criminal law and the physicians ethical code with respect to saving lives could be discussed more deeply in the discussion section. 

Author Response

Thank you to the reviewer for the time and insightful feedback. We are also very grateful for the appreciation of our study. As suggested, we have revised Table 1 accordingly to the indications.

Further, we added sub-chapter 4.5. “Contrasting values” in the “Debated issues” paragraph, for a discussion of the main arguments on ‘pros’ and ‘cons’ of E/PAS legitimation, indicating the positions also of several medical associations.

We also tried to highlight the main bioethical concepts on the subject, trying to summarize them given the length of our paper, which otherwise could become unreadable.

Anyway, we considered it useful to report an essential bibliography for further analysis.

We hope to have satisfied the reviewer’s requests with our changes.

Best regards, sincerely

Round 2

Reviewer 2 Report (New Reviewer)

The authors have answered sufficiently to the review questions. 

This manuscript is a resubmission of an earlier submission. The following is a list of the peer review reports and author responses from that submission.

Round 1

Reviewer 1 Report

- I do not see a consistent structure in the paper...11111?

- why underlining page 3 and 4?

- where are the figures, mentioned on p. 2?

-  concerning Belgium: why are the reports from different dates?

- is the interpretation of Kant on p. 9 correct? I doubt it.

These are quite a lot of interesting informations, but the way how they are structured needs amelioration.  

Author Response

We would like to thank the reviewer for the time and insightful feedback. We are also greatful for the appreciation of the study. Based on the suggestions provided, we have tried to modify the paper.

Please see our responses below and in the revised manuscript.

“- I do not see a consistent structure in the paper...11111?”

“- why underlining page 3 and 4?”

Thanks for the advice. We proceeded to fix the structure of the paper. Previously, the errors were probably due to the automatic generation from the website.

“- where are the figures, mentioned on p. 2?”

We apologize to the reviewer. Indeed, we had proceeded to elaborate some figures which, however, due to technical problems, we decided to eliminate. We fixed the paper.

“-  concerning Belgium: why are the reports from different dates?”

Thank you, this is an interesting point and we were actually doubtful about whether to report only the most recent data. Ultimately, we chose to report the data from different dates to highlight the chronological trend of E/PAS cases, in general, and for mental disorders. This information may be interesting considering the "slippery slope" argument, which we then analyzed later in a specific sub-chapter. Further, we decided to maintain the data from 2003 to 2013 as they are more detailed and rich in information than the recent ones, which probably still need to be reviewed.

“- is the interpretation of Kant on p. 9 correct? I doubt it.”

We thank the reviewer for the opportunity to discuss this topic. We based our sentence on Dietmar von der Pfordten's writing - "Considerations on the concept of human dignity" - which unfortunately we only found in Italian. However, let us try to make a brief summary of the study, in relation to our sentence.

In his article, the Author underlines the precariousness of the qualification of dignity in a “performance" sense. So, a distinction is drawn between “contingent” dignity and “necessary” (“inherent”) dignity, as a quality inherent in all human beings.

The Author recalls Kant's definition when he defines "dignity" as the quality of a rational being "who obeys no other law than that which he himself at the same time he gives to himself" (I. KANT, Grundlegung, cit., p. 434.).

Kant excludes an ultimate reconciliation of "being an end in oneself" to other normative sources, i.e., to sources beyond the individuals themselves, who belong, as rational beings, to a common Kingdom of Ends.

Dignity is characterized as an explication of the "condition" of being "an end in oneself", not as its direct manifestation. Elsewhere Kant writes: "Autonomy (Selbstgesetzgebung) is therefore the foundation of dignity of human nature and of every rational nature" (I. KANT, Grundlegung, cit., p. 435).

Human dignity is not based on an intersubjective recognition and It must be interpreted, on the contrary, in an internal or individual sense, where autonomy is the foundation.

In conclusion, individuals represent the ultimate ethical authority and only they can in principle decide independently about justifiable qualities, of which the first and foremost interest consists of the desire or second-order interest in having primary interests (these may be goals, desires, needs, or impulses). While needs and impulses are nevertheless unintentional, desires and goals are iterable and establish self-determination.

Our sentence derives precisely from this last Kantian conception.

PS: Finally, as suggested, we performed a review of the English language and style.

Reviewer 2 Report

Assisted suicide and euthanasia in mental disorders are issues involving challenges at both professional and legislative level.  Perhaps more information on decision-making and professional ethics could be integrated when working in multidisciplinary teams.

The article presents a systematic review of countries that have legislated on these issues. This is relevant as the number of psychiatric patients requesting Assisted suicide and euthanasia continues to increase. I suggest that you can add the process for the literature search and databases used.

I found several articles on euthanasia and assisted suicide, but very few where ethical principles in relationship with proportionality, dignity, and the right to die were analyzed.

I think it is a very interesting article, which presents an analysis based on evidence

Author Response

Thank you to the reviewer for the time and insightful feedback. We are also very grateful for the appreciation of our study. As suggested, we have added information about the process for the literature search and databases used. We added this information in the section “Data Availability Statement” of the manuscript, as it seemed to us a more appropriate solution in order not to reduce fluidity in reading the text.

Finally, as suggested, we performed a check of the English language and style.

Reviewer 3 Report

Overall the review describes a controversial topic regarding the ethical and legal implications for assisted suicide and euthanasia in mental disorders.

The authors overarching aim is to provide a global perspective on this complex issue which considers clinical practice, the irreversibility of the disease, assessment of the state of physical and mental suffering, as well as the possibility of adopting free and informed choices. This is an interesting, albeit controversial stance on euthanasia and peer assisted suicide for patients who may not be able to make a rational, nor informed choice based on their psychiatric state. 

 A few points to consider below:

·       ‘Mental disorders’ is an extremely broad term.  It essentially includes anything from depression, to autism and dementia.  If you include age and gender, it complicates everything further.  As such, I believe it is very difficult to draw overall conclusions, let alone global recommendations.

·       Similarly, the authors endeavour to take into consideration many sub topics including ethics, tradition, patriarchal approaches to healthcare, religion, dignity, and the right to die.   The review should be more focussed.

·       I don’t think it’s appropriate to have a position statement based on the review of the literature.

Author Response

We would like to thank the reviewer for the time and insightful feedback. We also thank the reviewer for the acknowledgment that our paper examines a controversial and debated topic.

Based on the suggestions provided, we have tried to modify the paper.

Please see our responses below and in the revised manuscript.

- “·‘Mental disorders’ is an extremely broad term.  It essentially includes anything from depression, to autism and dementia.  If you include age and gender, it complicates everything further.  As such, I believe it is very difficult to draw overall conclusions, let alone global recommendations.”

This is a good point and we thank the reviewer for the opportunity to clarify our intention. Our main purpose was to provide a broad consideration of the topic of E/PAS in subjects without exclusively somatic diseases. We have chosen not to focus only on subjects with psychiatric disorders or, alternatively, with neurocognitive disorders, as we have found the presence in the literature of already very detailed studies on these specific topics. Therefore, we have tried to develop the E/PAS issue in general for people with mental disorders who may be burdened with limitations in self-determination and legal competence. In fact, people with severe mental disorders may share neurocognitive impairments in the same way as other people with primary cognitive disorders. We added such clarification at the end of the “Introduction” chapter.

- "Similarly, the authors endeavour to take into consideration many sub topics including ethics, tradition, patriarchal approaches to healthcare, religion, dignity, and the right to die.   The review should be more focussed."

About this consideration, recalling what was said before, we have tried to give different considerations from several points of view, deliberately without addressing a specific focus. As interested above all in medico-legal aspects, we have tried to provide comprehensive information on a subject that is highly controversial. As mentioned, in fact, many other Authors have written on the subject, each considering a specific area of interest and research. Our intent was to write a paper that collected all this information. Moreover, we have also tried to provide cultural and religious aspects to highlight the differences and controversies also from these perspectives.

- "I don’t think it’s appropriate to have a position statement based on the review of the literature.”

Thank you to the reviewer for this suggestion. We agree that giving a position statement could be a hazard. We have therefore changed the text. However, we have tried to perform our reflections not only on what emerged from the literature review but also on aspects acquired in bioethical and medico-legal fields, or involving correlated aspects. In this sense, we hope to have provided reflections with a theoretical foundation.

Finally, as suggested, we performed a check of the English language and style.

Round 2

Reviewer 1 Report

Dear Authors, many thanks for the adaptations of your contribution...this makes a great difference!

One comment: in my view, there is not a right to health, but to health care (health can not be guaranteed).